

# DARUMA: a gateway to fast and easy prediction of intrinsically disordered regions

Itsuki Shimizu, Takuya Ida, Yuhei Ozawa, Satoshi Fukuchi and Hiroto Anbo

Division of Informatics, Bioengineering and Bioscience, Faculty of Engineering, Maebashi Institute of Technology, Maebashi, Gunma, Japan

## ABSTRACT

**Background:** Intrinsically disordered proteins (IDPs) are proteins that contain intrinsically disordered regions (IDRs), which lack stable three-dimensional structures under physiological conditions. These regions are known to play crucial roles in many biological processes. While IDRs can be predicted from their amino acid sequences, and several accurate IDR prediction programs have been developed, such programs often require substantial computational resources, including long execution times, large databases for homology searches, and advanced computer architectures. Since DNA sequence data continues to grow rapidly, particularly at a genomic scale, there is an increasing need for fast and accurate IDR prediction programs that demand fewer computational resources.

**Methods:** In this study, we developed DARUMA (Disorder order clAssifier by Rapid and User-friendly MAchine), an IDR prediction program designed for speed and ease of use. DARUMA uses a one-dimensional convolutional neural network (1D-CNN) that processes the physicochemical properties of amino acid residues instead of relying on sequence profiles. DARUMA employs a simple neural network that predicts IDRs using the output of 1D-CNN as input features. To ensure easy installation on users' systems, DARUMA was written entirely in Python using standard and NumPy libraries.

**Results:** DARUMA achieves fast performance by avoiding iterative homology searches while delivering accuracy comparable to the latest predictors that use sequence profiles. In addition to the advantage of execution time, DARUMA requires no additional homology search programs and operates using standard Python libraries, making it easy to install and run on users' own environments without the need for specialized computational resources. DARUMA is available at https://antepontem.org/daruma/, which also provides the stand-alone distribution.

## INTRODUCTION

Intrinsically disordered proteins (IDPs) have attracted much attention in protein science over the past two decades (*Uversky, 2002*; *Wright & Dyson, 1999*). IDPs are proteins that contain intrinsically disordered regions (IDRs), which do not adopt stable 3D structures

Corresponding author
Hiroto Anbo,
hanbo@maebashi-it.ac.jp

under physiological conditions (*Dunker et al., 2001*). Despite lacking a stable structure, IDRs play key roles in essential biological processes, such as transcriptional regulation and signal transduction, challenging the traditional concept that proteins function solely by forming stable three-dimensional structures (*Dyson & Wright, 2005*). Following the discovery of IDPs, one of the major breakthroughs in molecular biology has been the development of sequencing techniques (*Shendure & Ji, 2008*). Advances in next-generation sequencing have led to an extraordinary surge in DNA sequence data. In 2002, GenBank housed approximately 17 million sequences in its main section and 170,000 sequences in the whole genome shotgun (WGS) section. In 2024, these numbers had grown to 250 million in GenBank and 3.4 billion in WGS (*NCBI, 2025*) (https://www.ncbi.nlm.nih. gov/genbank/statistics/). Since GenBank primarily stores sequence data for individual or small groups of genes, while the WGS section contains data from whole genome shotgun analyses, this trend indicates a shift from sequencing individual genes to genome wide sequencing. Reflecting the trend, UniProtKB provides 843,388 proteome datasets on March 8th, 2025 (*UniProt, 2025*) (https://www.uniprot.org/).

IDR prediction programs have played a crucial role in shaping our understanding of IDPs since the early stages of IDP research. In particular, proteome-scale predictions of IDRs have enabled comparisons of IDR ratios between model organisms and facilitated analyses of sub-cellular localizations, post-translational modifications, and protein-protein interactions in IDPs (*Dunker et al., 2008*; *Fukuchi et al., 2011*; *Haynes et al., 2006*; *Iakoucheva et al., 2004*; *Minezaki et al., 2006*; *Patil & Nakamura, 2006*; *Ward et al., 2004*). Such research has suggested that eukaryotic genomes encode a substantial number of IDPs, with 30–40% of residues in eukaryotic proteomes found in IDRs. Additionally, IDPs are abundant in nuclear proteins and tend to serve as hub proteins with more interaction partners in protein-protein interaction networks. These insights have become widely accepted in IDP research, largely based on computational predictions of IDRs (*Oldfield & Dunker, 2014*). As a result, experimental determination of IDRs is seldom conducted for annotation purposes, with IDR predictions typically being accepted as annotations. Given the rapid growth in proteome-scale sequence data and the advantages of large-scale IDR analysis, the demand for fast and accurate IDR prediction programs continues to rise.

Over the past two decades, numerous IDR predictors have been developed and evaluated in competitions such as CASP5 to CASP10 (*Moult et al., 2014*; *Moult et al., 2003*) and the Critical Assessment of protein Intrinsic Disorder prediction (CAID) (*Conte et al., 2023*; *Necci et al., 2021*). Typically, IDR prediction involves converting amino acid sequences into feature values, such as the physicochemical properties of amino acid residues, their secondary structure propensities, and/or sequence profiles (*He et al., 2009*; *Meng, Uversky & Kurgan, 2017*). Although there are many disorder predictors, the CAID2 challenge found that the predictors using sequence profile have the advantage in the disorder prediction accuracy (*Conte et al., 2023*). Sequence profiles are created from multiple sequence alignments, which are generally generated by iterating homology searches against a reference database. Most profile driven IDR predictors rely on position-specific scoring matrices (PSSM), a type of sequence profile produced by PSI-BLAST (*Altschul et al., 1997*). While sequence profile-driven predictors generally

achieve high accuracy, they come with significant computational costs due to resource-intensive process of generating sequence profiles. Homology search programs typically require a large sequence database as the search target and their iterative processes to identify remote homologs are time consuming (*Altschul et al., 1997*).

IDR prediction methods are roughly divided into three categories: machine learning methods, scoring function methods, and consensus methods. The accumulation of IDR annotations in IDP databases (*Fukuchi et al., 2014*; *Hatos et al., 2020*; *Piovesan et al., 2018*), along with advances in machine learning, has enabled the development of more powerful prediction models. Machine learning-based predictors typically provide highly accurate predictions but require high-spec hardware, including large amounts of random-access memory (RAM) and multiple graphics processing units (GPUs). Additionally, some machine learning-based predictors rely on extra libraries and packages, making their installations on local environments challenging. Given the rapid growth of sequence data, particularly proteome data, and the increasing demand for IDR annotations, a cost-effective, user-friendly program that maintains high accuracy is highly desirable.

In this study, we developed a fast, accurate, and user-friendly predictor called DARUMA (Disorder order clAssifier by Rapid and User-friendly MAchine). DARUMA uses a convolutional neural network (CNN) (*Lecun et al., 1998*) that processes the physicochemical properties of amino acid residues instead of relying on sequence profiles. Written entirely in Python using standard and NumPy libraries, DARUMA's design ensures easy installation on users' systems. Its streamlined specification allows for rapid predictions, making it especially suitable for large-scale analyses.

# MATERIALS AND METHODS

## Feature value embedding

To represent the features of amino acid sequences, we used the AAindex database (*Kawashima & Kanehisa, 2000*), which provides 20 numerical values for various physicochemical and biological properties of amino acids. The AAindex database is divided into three sections, AAindex1, AAindex2, and AAindex3 (*Kawashima et al., 2008*). We selected AAindex1, which contains 566 indexes for the physicochemical properties of each amino acid, including factors such as side chain size and hydrophobicity. We utilized 553 indexes as feature values, excluding all those with N/A values. The values were normalized to a scale from 0 to 1 across the 20 numerical properties. Consequently, each amino acid residue was represented by a 553-dimensional vector of physicochemical properties, and an amino acid sequence with L residues was embedded into an $L \times 553$-dimensional vector.

## Construction of training and validation datasets

To construct the training and validation datasets, we used two sources: the DM4229 dataset and the IDEAL database (*Fukuchi et al., 2014*) (https://www.ideal-db.org/). DM4229, which was originally used to develop the SPINE D program (*Zhang et al., 2012*), provides IDR annotations for 4,229 proteins. IDEAL, an IDP database, offers IDR

annotations for 981 proteins. To eliminate redundancy between DM4229 and IDEAL, we performed clustering using BLASTClust (*Altschul et al., 1997*) with a 25% identity threshold, resulting in 4,981 clusters containing a total of 5,210 proteins. We selected representative proteins from each cluster as follows: the longest protein from IDEAL if a cluster contained multiple IDEAL proteins, an IDEAL protein if the cluster had one IDEAL protein, and the longest protein if a cluster had no IDEAL protein. IDEAL proteins were prioritized over those from DM4229 because IDEAL is continuously updated (*IDEAL, 2025*) (https://www.ideal-db.org/history.html). The CAID project, which evaluates IDR prediction programs, provides datasets for performance assessment. Since we used the CAID dataset as evaluation dataset, we excluded proteins with 25% or greater similarity to any CAID proteins from the representative proteins. This resulted in a final set of 4,762 proteins. Of these, 4,286 proteins (3,731 from DM4229 and 555 from IDEAL; Table S1) were assigned to the training dataset, and 476 proteins (414 from DM4229 and 62 from IDEAL; Table S1) were used for the validation dataset. To further enhance generalization ability, we included another validation dataset, the SL dataset (https://mendel.bii.a-star.edu.sg/SEQUENCES/disorder/), provided by *Sirota et al. (2010)* as a benchmark set for IDR predictor. We removed redundant proteins with 25% sequence identity found in the other datasets from the SL dataset.

## Construction of evaluation dataset

We used the dataset from the first round of CAID (CAID1) (*Necci et al., 2021*), DisProt646 and DisProt-PDB646 (https://codeocean.com/capsule/2223745/tree/v1), for evaluation. While both DisProt646 and DisProt-PDB646 contains 646 proteins from the DisProt database (*Hatos et al., 2020*), they follow different policies for disorder annotation. In both datasets, a residue is labeled as positive if it has a disorder annotation in DisProt. However, DisProt646 labels a residue as negative if it lacks a disorder annotation in DisProt, whereas DisProt-PDB646 labels a residue as negative only if it lacks a disorder annotation in DisProt and is part of a PDB structure. Consequently, DisProt-PDB646 includes residues labeled as neither positive nor negative (undefined in Table S1), whereas DisProt646 classifies all residues as either positive or negative. For performance evaluations using DisProt-PDB646, undefined residues were excluded. Additionally, we evaluated our method using the two datasets from the second round of CAID (CAID2) (*Conte et al., 2023*): disorder_nox and disorder_pdb (https://caid.idpcentral.org/challenge/results). Redundant proteins shared with the training and validation datasets were removed from these evaluation sets. The disorder_pdb dataset follows the same labeling policies as DisProt-PDB646. In contrast, disorder_nox labels a residue as positive if it is identified as disordered by circular dichroism and registered in DisProt, with all other residues labeled as negative. The detailed statistics of these datasets are provided in Table S1.

## DARUMA architecture

DARUMA comprises two units: a feature extraction unit (FEU) and a prediction unit (PU) (Fig. 1). The FEU employs a CNN featuring three 1D convolutional layers (*Lecun et al., 1998*). A query sequence is embedded into an $L \times 553$ vector, which is then zero-padded

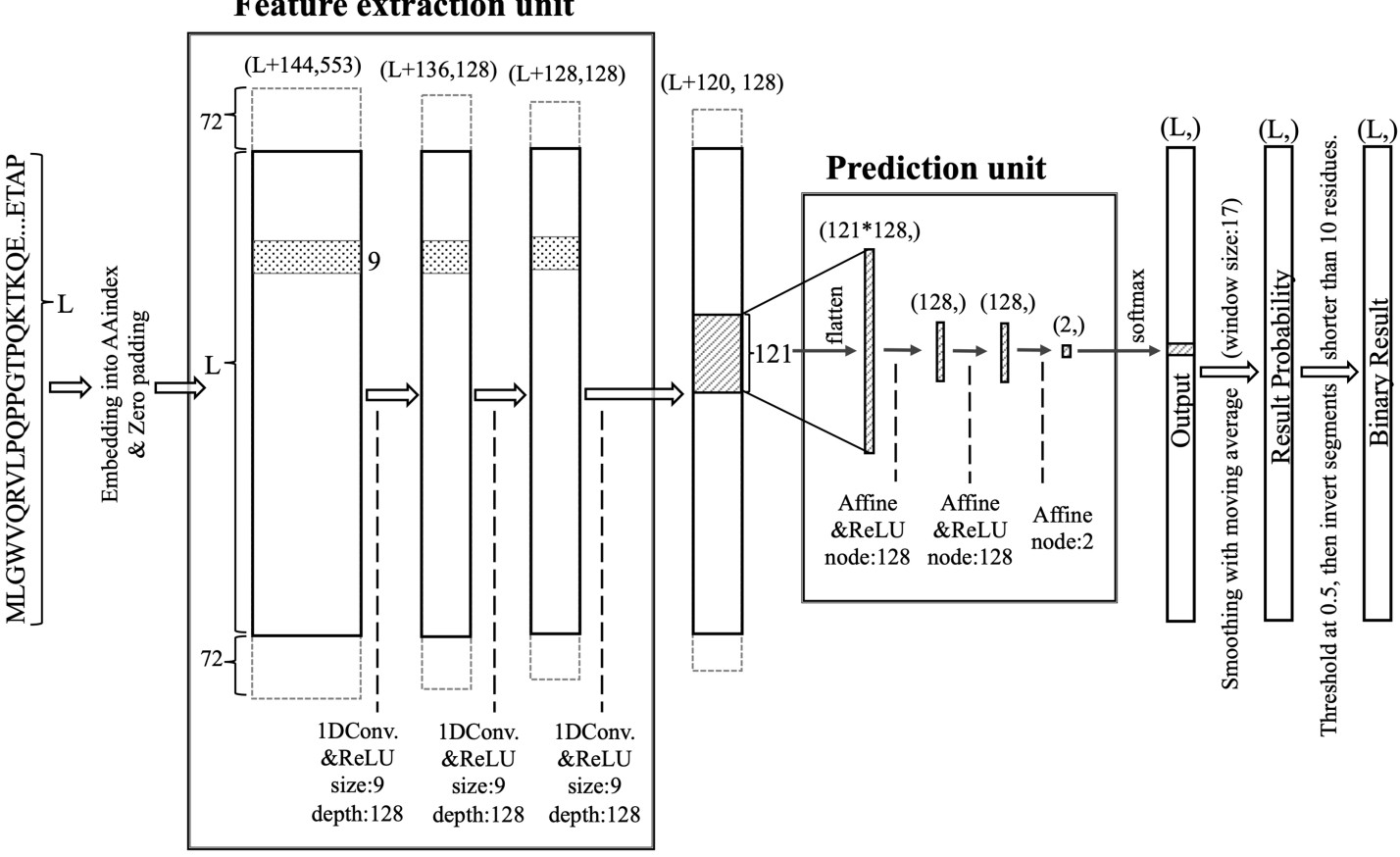

**Figure 1 Structure of the DARUMA model.** L represents the length of the query sequence. The values in parentheses indicate the dimensions of each layer. The values added to L at the first dimension in parentheses are the size of the zero-padded region. The boxes, flanking the solid-lined box and surrounded by dashed lines, indicate the zero-padded region. The figures associated with size and depth denote the size and number of filters, respectively. The term node refers to the number of nodes that make up the affine layer.

with 72 residues on both terminals before inputting into the first 1D convolutional layer. The filter size and the number of filters for all 1D convolutional layers are set to 9 and 128, respectively. Each layer uses the rectified linear unit (ReLU) (*Hahnloser et al., 2000*; *Nair & Hinton, 2010*) activation function. The vector size decreases by eight residues at each 1D convolutional layer since the padding layer is not utilized in the FEU. As a result, the output from the final 1D convolutional layer consists of feature values compressed into an (L + 120) × 128-dimensional shape, which are then processed by the PU after applying the sliding window technique. In this sliding window process, the window size and the sliding step size are set to 121 and 1, respectively. The PU utilizes a neural network (*McCulloch & Pitts, 1990*; *Rosenblatt, 1958*) that includes an input layer with 121 × 128 nodes, two hidden layers each containing 128 nodes, and an output layer with 2 nodes. The ReLU activation function is applied to the hidden layers, while the output layer employs the Softmax activation function (*Bridle, 1990*). The output from the Softmax node is averaged over 17 residues centered around a residue of interest; if the average exceeds 0.5, that residue is

classified as disordered. When the 17-window was not applicable at the terminal regions, values over all residues were averaged. Additionally, IDRs shorter than 10 residues are relabeled as structured, while structured regions shorter than 10 residues are reclassified as disordered.

## Optimization of DARUMA

We optimized all hyperparameters in DARUMA through a grid search, selecting those that demonstrated the highest average performance across the two validation datasets. Additionally, we optimized the size of the sliding window applied to the outputs of the Softmax node. A detailed description of the hyperparameter selection process is provided in the additional information.

## Evaluation of DARUMA

We selected eight IDR predictors as reference programs: IUPred3-long, IUPred3-short (*Erdos, Pajkos & Dosztanyi, 2021*), SETH (*Ilzhofer, Heinzinger & Rost, 2022*), SPOT-Disorder-Single (*Hanson, Paliwal & Zhou, 2018*), SPOT-Disorder (*Hanson et al., 2017*), SPOT-Disorder2 (*Hanson et al., 2019*), flDPnn (*Hu et al., 2021*), and rawMSA (*Mirabello & Wallner, 2019*). IUPred3-long, IUPred3-short, SETH, and SPOT-Disorder-Single do not utilize sequence profiles, and we categorized them as profile-free predictors. In contrast, SPOT-Disorder, SPOT-Disorder2, and flDPnn use sequence profiles and are classified as profile-driven predictors. Since rawMSA uses homology searching, we categorize it as the profile-driven predictor, although it utilizes raw multiple sequence alignments without generating profiles. Among these, SETH, rawMSA, SPOT-Disorder-Single, SPOT-Disorder, SPOT-Disorder2, and flDPnn employ machine learning methods, with the first two utilizing CNN, the following three utilizing long short-term memory methods (*Hochreiter & Schmidhuber, 1997*), and flDPnn employing a deep neural network (*Hinton, Osindero & Teh, 2006*) consisting of two fully-connected layers, all as their main model architecture. IUPred3-long and -short use a statistical potential to describe intra-chain contacting residue pairs, resulting in shorter execution times compared to profile-driven predictors (*Meszaros, Erdos & Dosztanyi, 2018*). To ensure fair comparisons of execution time, all predictors were run on a machine equipped with an Intel Xeon E5-2697 v4 processor at 2.30GHz and 128GB of RAM, without the use of a GPU. The execution time was normalized by sequence length as follows: total execution time across all proteins in a dataset was divided by the total number of residues and multiplied by 1,000.

To measure the performance of the predictors, we utilized the Matthews Correlation Coefficient (MCC) (*Matthews, 1975*), the area under the curve of receiver operating characteristics (AUCroc) (*Hanley & McNeil, 1982*), sensitivity, and precision.

MCC, sensitivity, and precision are represented as follows:

$$MCC = \frac{TP \times TN - FP \times FN}{\sqrt{(TP + FP)(TP + FN)(TN + FP)(TN + FN)}},$$

$$Sensitivity = \frac{TP}{TP + FN}, \text{ and}$$

$$Precision = \frac{TP}{TP + FP}$$

where TP, TN, FP, and FN represent true positives, true negatives, false positives, and false negatives, respectively. We utilized the default thresholds from the reference predictors to differentiate between positive and negative. Portions of this text were previously published as part of a preprint (https://www.researchsquare.com/article/rs-5414158/v1).

## RESULTS

We compared the performance of DARUMA with that of other predictors. Table 1A presents the evaluation results using the CAID1 dataset, DisProt646, and DisProt-PDB646. Among the profile-free programs, DARUMA and SETH, which utilize CNN, achieved the highest performance in MCC, and DARUMA, SETH, and SPOT-Disorder-Single exhibited a comparable performance in AUCroc for the DisProt646 dataset. Overall, the profile-driven predictors achieved high performances, flDPnn especially showed notable performances in both MCC and AUCroc. A similar trend is observed for the DisProt-PDB646 dataset, where the profile-driven predictors also demonstrated good performance. DARUMA excelled among the profile-free predictors and outperformed flDPnn and rawMSA, which are the profile-driven predictors, in both MCC and AUCroc. Interestingly, although flDPnn achieved high performance in the DisProt646 results, it recorded the lowest accuracy. Table 1B presents the evaluation on the CAID2 dataset. Consistent with the findings from DisProt646 and DisProt-PDB646, DARUMA performed well among the profile-free predictors, achieving the highest MCC for both datasets and the highest AUCroc for disorder_pdb. Overall, among all predictors, including the profile-driven ones, DARUMA ranked second in MCC for the disorder_nox dataset and the third in MCC and AUCroc for the disorder_pdb dataset, demonstrating comparable performance of the profile-driven predictors. Those results show that DARUMA exhibited consistent performances across all test datasets, indicating its robustness in IDR prediction.

Next, we compared the execution times of the predictors on the CAID1 dataset. To determine the execution time for each predictor, we calculated the execution time per 1,000 residues across all proteins in the datasets. Figure 2 shows a scatterplot illustrating the relationship between execution time and MCC. In both datasets, DARUMA, IUPred3-long and -short emerged as the fastest predictors. Among these, DARUMA showed comparable performance to the profile-driven predictors in both datasets. Although SETH showed high performance along with DARUMA, it took 30.19 s, which was 33.5 times longer than DARUMA. Generally, profile-driven predictors yield good performance but require more time to execute (Conte et al., 2023). This trend is evident in Fig. 2, particularly in the case of SPOT-Disorder2. flDPnn was the fastest among the profile-driven predictors,

**Table 1 Performance metrics for of all predictors evaluated on the CAID1 and CAID2 datasets.**

**(A) Performance metrics on the CAID1 dataset**

| | | DisProt646 | | DisProt-PDB646 | |
|---|---|---|---|---|---|
| | | MCC | AUCroc | MCC | AUCroc |
| Profile-Free | DARUMA | 0.315 | 0.753 | 0.678 | 0.910 |
| | SPOT-Disorder-Single | 0.278 | 0.757 | 0.614 | 0.897 |
| | IUPred3-long | 0.286 | 0.738 | 0.591 | 0.865 |
| | IUPred3-short | 0.278 | 0.742 | 0.562 | 0.861 |
| | SETH | 0.329 | 0.756 | 0.689 | 0.894 |
| Profile-Driven | SPOT-Disorder | 0.309 | 0.750 | 0.698 | 0.917 |
| | SPOT-Disorder2 | 0.332 | 0.765 | 0.711 | 0.924 |
| | flDPnn | 0.361 | 0.814 | 0.534 | 0.874 |
| | rawMSA | 0.327 | 0.785 | 0.639 | 0.902 |

**(B) Performance metrics on the CAID2 dataset**

| | | disorder_nox | | disorder_pdb | |
|---|---|---|---|---|---|
| | | MCC | AUCroc | MCC | AUCroc |
| Profile-Free | DARUMA | 0.420 | 0.792 | 0.738 | 0.923 |
| | SPOT-Disorder-Single | 0.371 | 0.793 | 0.668 | 0.917 |
| | IUPred3-long | 0.377 | 0.774 | 0.656 | 0.885 |
| | IUPred3-short | 0.332 | 0.763 | 0.600 | 0.873 |
| | SETH | 0.401 | 0.793 | 0.711 | 0.911 |
| Profile-Driven | SPOT-Disorder | 0.416 | 0.799 | 0.754 | 0.930 |
| | SPOT-Disorder2 | 0.452 | 0.786 | 0.794 | 0.949 |
| | flDPnn | 0.411 | 0.840 | 0.544 | 0.893 |
| | rawMSA | 0.345 | 0.798 | 0.634 | 0.910 |

taking 47.13 s, which is about 52 times longer than DARUMA. Overall, DARUMA can deliver high-speed IDR predictions with accuracy comparable to that of the profile-driven predictors.

DARUMA offers significant benefits to users, particularly for large-scale analyses. Figure 3 schematically illustrates the execution time distributions of IUPred3-long, IUPred3-short, and DARUMA. We utilized the DisProt646 dataset, which comprises 646 proteins, to calculate the execution time for each protein. The execution time are depicted in the boxplot. DARUMA(multi) represents the execution time distribution when a query is input in a multiple FASTA file containing a large number of amino acid sequences in FASTA format. When the number of sequences is N, DARUMA(single) presents the execution time distribution for a protein based on N independent executions of DARUMA. In these two processes, DARUMA(multi) performs the initialization process only once because the parallelization is not implemented even on DARUMA(multi), whereas DARUMA(single) executes the initiation N times. In the multiple FASTA query, the execution time per protein is 0.037 s, which is about 12.6 times faster than DARUMA (single) and 14.9 times faster than the two IUPred3 predictors. Additionally, the execution
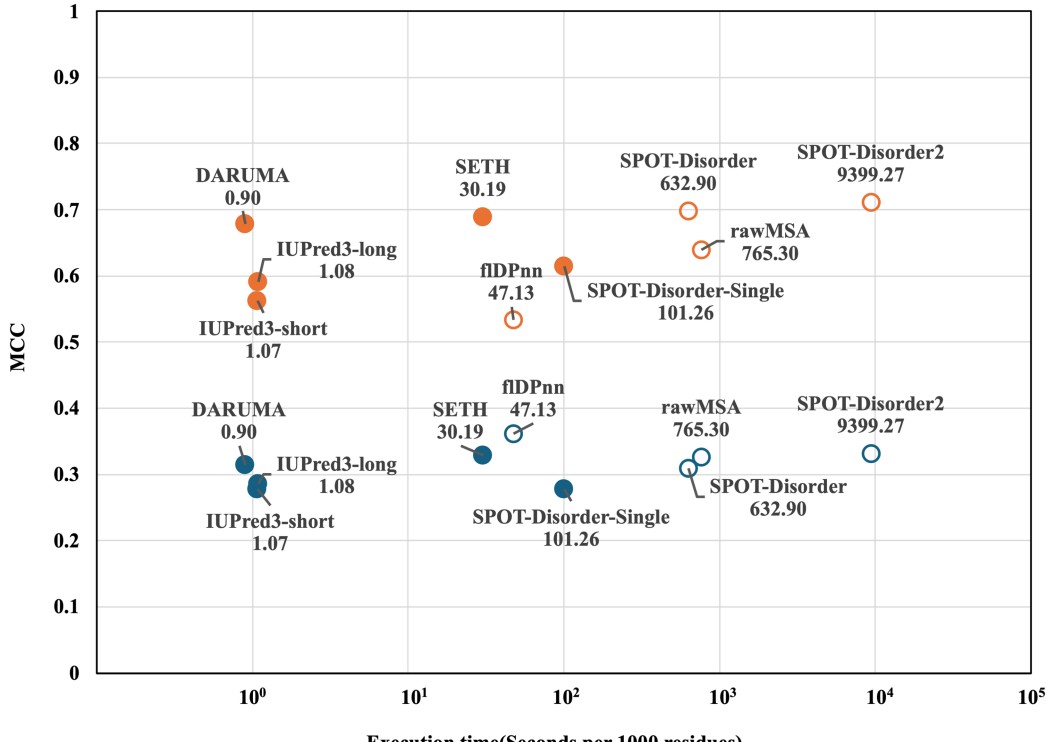

**Figure 2 MCC and execution time of each predictor on the two CAID1 datasets.** Execution time and MCC for each predictor are plotted on the horizontal and vertical axes. DisProt646 and DisProt-PDB646 are indicated as blue and orange, respectively. Filled and open circles indicate the profile-free predictors and profile-driven predictors, respectively. The names and execution times of the predictors are displayed alongside their corresponding markers. Please note that the horizontal axis is presented on a logarithm scale.

times of DARUMA(multi) and DARUMA(single) scaled linearly with sequence length, maintaining consistent differences across individual proteins (Fig. S1). This indicates that the initialization time of the prediction model accounts for majority of DARUMA's execution time. SETH can also accept a multiple FASTA file as an input; however, we could not include it in Fig. 3 because we could not calculate the execution time for each protein. DARUMA was 58.4 times faster than SETH in the total execution time for the DisProt646 dataset.

DARUMA is available through a web interface that supports two types of input: single sequence and multiple sequences (Fig. 4A). For a single FASTA input, DARUMA presents the results with a graphic chart, displaying the disorder probability for each residue in a line graph, while highlighting the disordered regions in red on the query sequence (Fig. 4B). In addition to the graphical outputs, users can download the results in text format by clicking the button in the upper right corner of Fig. 4B. When sequences are input in the multi FASTA format, users receive an e-mail containing a URL that leads to a results list (Fig. 4C). Additionally, DARUMA is provided as a Python pip package, within a Docker container, and through GitHub for easy installation in local environments. These services simplify the installation process, allowing DARUMA to run on both Python2 and

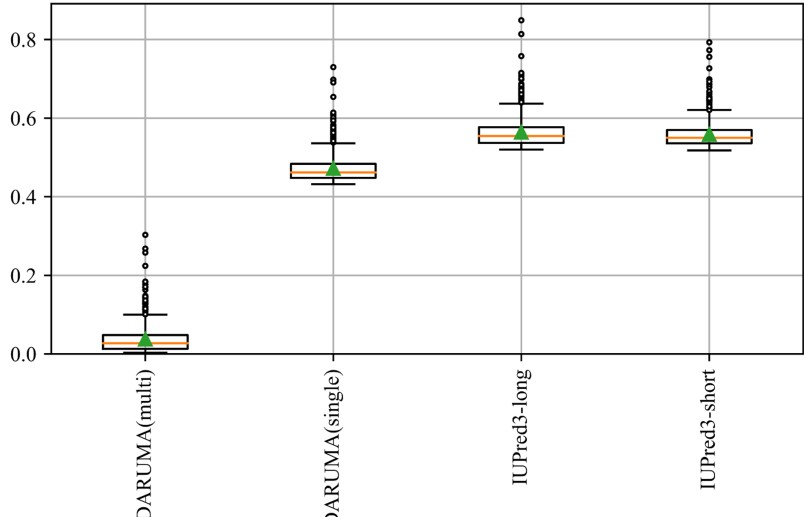

**Figure 3 Execution time of each predictor on the DisProt646 dataset.** The boxplot provides a schematic representation of the execution time distribution for proteins in the DisProt646 dataset. The green triangle and orange line indicates the average and median execution time, respectively. The box represents the quartile range, horizontal bar depicts the full range of the dataset, and circles denote outliers. DARUMA(single) and DARUMA(multi) represent the execution time for a single protein query and a query in the multi FASTA format.              

Python3 with only the standard and NumPy libraries required. Installation packages can be found at https://antepontem.org/daruma/download.html.

## DISCUSSION

DARUMA achieved a high MCC and area under the curve (AUC) (Table 1) among the profile-free predictors due to its high sensitivity of 0.718 (Table 2). Since the definitions of positive in DisProt646 and DisProt-PDB646 are the same, the sensitivity, true positives, and false negatives are identical between these two datasets. Generally, as the number of residues predicted as positive increases, the number of false positives (FPs) also rises, conversely, an increase in negative predictions typically leads to a higher count of false negatives (FNs). In fact, DARUMA had the second-highest number of true positives (TPs) and the third-highest number of false positives, while flDPnn, which recorded the highest MCC and AUCroc (0.361 and 0.814 in Table 1A) in DisProt646, had the highest number of true negatives (TNs) and FNs. These results indicate that DARUMA tends to yield more positive predictions, whereas flDPnn generates more negative predictions. SPOT-Disorder and SPOT-Disorder2, which are the profile-driven predictors, and SETH, which is the profile-free predictors, display a similar trend to DARUMA. Considering the trade-off between the numbers of positive predictions and the risk of misclassifications, it is important for users to understand the characteristics of each predictor.

In the evaluation using DisProt-PDB646, the accuracy of all predictors improved compared to their performance with DisProt646. In DisProt646, residues with disordered annotations are labeled as positive, while others are labeled as negative. In contrast, in DisProt-PDB646, only residues with ordered annotations are labeled negative, with the

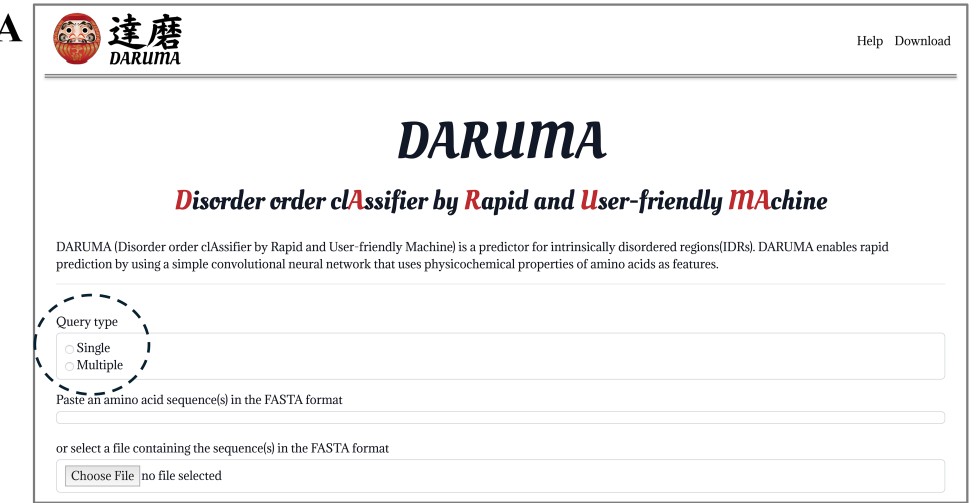

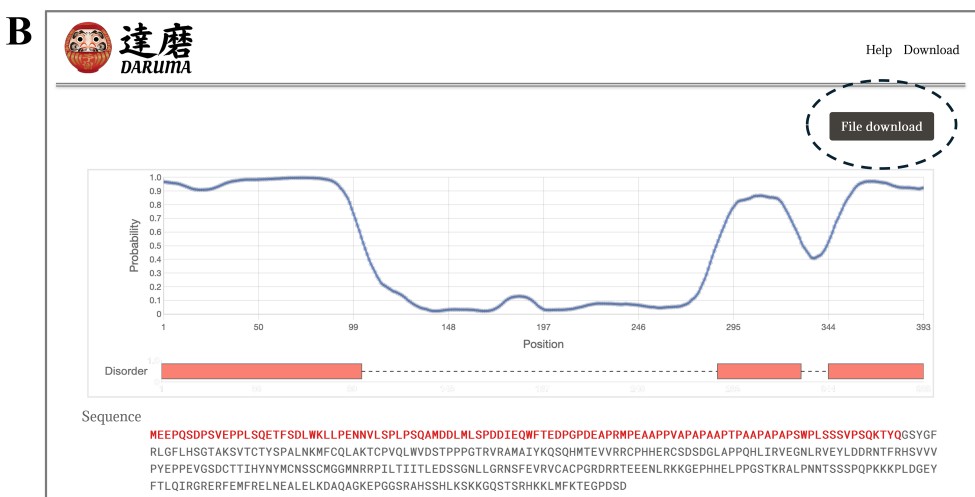

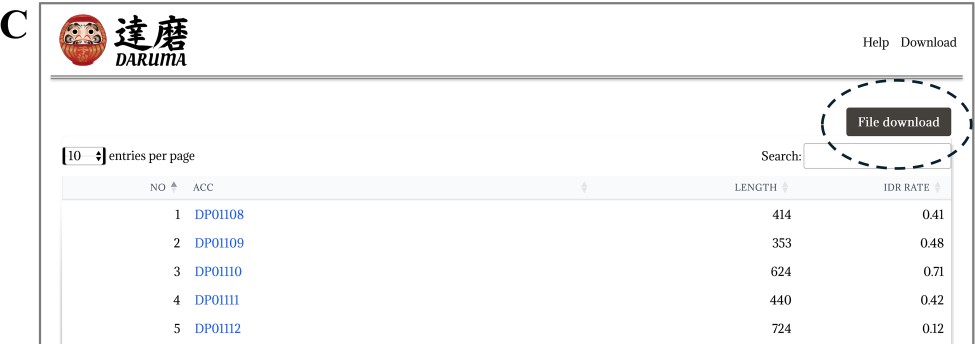

**Figure 4 Web interface of DARUMA.** (A) The submission page of DARUMA. The dotted circle highlights the button for selecting either single FASTA or multiple FASTA formats. Users can enter a sequence or sequences in the box below, or they can upload a file. (B) An example of the DARUMA results. The blue line graph schematically displays the disordered probabilities along the query sequence, while the orange box indicates the disordered regions based on binary predictions. The button enclosed by the dotted circle allows users to download the results in text format. (C) The list of results submitted *via* a multi FASTA file. Users receive an e-mail containing the URL for this page. The blue letters are hyperlinks to the results formatted as shown in panel B. The button enclosed by the dotted circle enables users to download an archived file containing each result in text format.

**Table 2  Additional performance metrics for all predictors on the DisProt646 and DisProt-PDB646 datasets.**

| | | Sensitivity | Precision | TP | TN | FP | FN |
|---|---|---|---|---|---|---|---|
| **DisProt646** | | | | | | | |
| Profile-Free | DARUMA | 0.718 | 0.313 | 39,182 | 196,163 | 85,828 | 15,422 |
| | SPOT-Disorder-Single | 0.556 | 0.329 | 30,384 | 220,025 | 61,966 | 24,220 |
| | IUPred3-long | 0.638 | 0.312 | 34,829 | 205,135 | 76,856 | 19,775 |
| | IUPred3-short | 0.557 | 0.329 | 30,415 | 219,870 | 62,121 | 24,189 |
| | SETH | 0.770 | 0.310 | 42,041 | 188,399 | 93,592 | 12,563 |
| Profile-Driven | SPOT-Disorder | 0.713 | 0.309 | 38,685 | 195,091 | 86,358 | 15,608 |
| | SPOT-Disorder2 | 0.742 | 0.331 | 36,651 | 161,985 | 73,967 | 12,777 |
| | flDPnn | 0.507 | 0.439 | 27,667 | 246,597 | 35,322 | 26,893 |
| | rawMSA | 0.660 | 0.340 | 36,034 | 211,959 | 70,032 | 18,570 |
| **DisProt-PDB646** | | | | | | | |
| Profile-Free | DARUMA | 0.718 | 0.826 | 39,182 | 114,203 | 8,263 | 15,422 |
| | SPOT-Disorder-Single | 0.556 | 0.887 | 30,384 | 118,590 | 3,876 | 24,220 |
| | IUPred3-long | 0.638 | 0.778 | 34,829 | 112,504 | 9,962 | 19,775 |
| | IUPred3-short | 0.557 | 0.807 | 30,415 | 115,209 | 7,257 | 24,189 |
| | SETH | 0.770 | 0.797 | 42,041 | 111,771 | 10,695 | 12,563 |
| Profile-Driven | SPOT-Disorder | 0.713 | 0.857 | 38,685 | 115,925 | 6,469 | 15,608 |
| | SPOT-Disorder2 | 0.742 | 0.847 | 36,651 | 106,382 | 6,610 | 12,777 |
| | flDPnn | 0.507 | 0.814 | 27,667 | 116,078 | 6,316 | 26,893 |
| | rawMSA | 0.660 | 0.823 | 36,034 | 114,725 | 7,741 | 18,570 |

remainder categorized as undefined. Therefore, the negative residues in DisProt646 are divided into negative and undefined categories in DisProt-PDB646. Since annotations are based on experimental evidence, the undefined residues in DisProt-PDB646 are those lacking any experimental validation. In this context, DisProt-PDB646 serves as a conservative dataset by excluding uncertain negative labels. These undefined residues may contain cryptic IDRs, and the residues in these regions can exhibit IDR properties. When a predictor identifies these residues as positive due to such IDR characteristics, this leads to false positives when evaluated against the DisProt646 dataset. The CAID experiment demonstrated that this characteristic of the dataset was one of the reasons for the increased accuracy (*Necci et al., 2021*).

While DARUMA, SETH, and rawMSA adopt CNN architectures for their prediction models, the input feature values differ among them. DARUMA, SETH and rawMSA utilize the physicochemical properties of each amino acid, the embedding values generated by the protein language model ProtT5, and raw multiple sequence alignments. Despite these differences, the overall performances of these predictors were comparable (Fig. 2).

Although we used eight predictors as reference predictors in the development of the DARUMA model, we compared the performance of DARUMA with 34 predictors available from CAID2 (https://caid.idpcentral.org/challenge/results) (Table S3). In order to ensure a fair comparison of execution time, we ran DARUMA on the Intel Xeon

**Table 3 Comparison of execution times across various computing configurations.**

| Configuration | Predictor | Minutes | Seconds per 1,000 residues |
|---|---|---|---|
| Xeon E5 128 G | IUPred3-long | 182.01 | 0.96 |
| | IUPred3-short | 180.13 | 0.95 |
| | DARUMA(multi) | 13.72 | 0.072 |
| Xeon Silver 92 G(on WEB) | DARUMA(multi) | 27.25 | 0.14 |
| Core i3 4 G 2nd Gen | DARUMA(multi) | 39.15 | 0.21 |
| Xeon Gold with GPU | DARUMA(multi) | 1.93 | 0.010 |

E5-2697 v4 processor at 2.30 GHz with RAM limited to 47 GB, which matches the RAM size used in the CAID2. Some profile-free predictors, such as SETH-0, metapredict, and DARUMA, show comparable performance, although the predictors demonstrating high performance were mostly profile-driven ones. DARUMA achieved high speed prediction than other two among these three. Figure S2 showed that DARUMA is a well-balanced IDR predictor in terms of both prediction accuracy and execution time.

We demonstrated in the results section that DARUMA can quickly process a file in the multi FASTA format. Fast predictions are particularly useful for large-scale analyses, which are often conducted on local computer environments. In such cases, execution time depends on the specifications of the running devices. Table 3 presents the execution time of four devices: the Intel Xeon E5-2697 v4 processor at 2.30 GHz with 128 GB of RAM (Xeon E5 128 G), the Intel Xeon Silver 4214R processor at 2.40 GHz with 92 GB of RAM (Xeon Silver 92 G), the Intel Core i3-2120 processor at 3.30 GHz with 4 GB of RAM (Core i3 4 G 2nd Gen), and the Intel(R) Xeon(R) Gold 5222 CPU @ 3.80GHz with GPU of RTX2080ti (Xeon Gold with GPU). Among these, the Xeon Silver 92 G is the device used for the web interface. The experiments were conducted using sequences from the human proteome, which includes 20,371 proteins (*UniProt, 2025*) (https://www.uniprot.org/). The execution times for processing the human proteome were 13.72, 27.25, 39.15 and 1.93 min on the Xeon E5 128 G, Xeon Silver 92 G, Core i3 4 G 2nd Gen, and Xeon Gold with GPU, respectively, resulting in per 1,000 residues execution times of 0.072, 0.14, 0.21 and, 0.010 s. Although the Xeon processors exhibited high performance in terms of speed, the Core i3 4 G 2nd Gen, a consumer-grade CPU released over a decade ago, demonstrated a practical execution time of less than an hour for the human proteome. The human proteome can be processed in less than 2 min by running DARUMA on a GPU. It shows that it is beneficial for large-scale analyses. These results indicate that DARUMA enables fast predictions on a wide range of processors, from consumer-grade to server-grade. Therefore, many users can run DARUMA in their respective environments.

Although we evaluated the performance of DARUMA with the programs locally installed in our environment, there are programs only available *via* the web interface. RIDAO (*Dayhoff & Uversky, 2022*) is the website to provide results of six disorder predictors and the consensus score. We could refer the execution time in its article (*Dayhoff & Uversky, 2022*), which can process about 420 million residues in 42 min due to multi-threaded execution. We calculated the execution time of DARUMA for the same

dataset, resulting in an execution time of 77.5 min. Since DARUMA conducts its prediction on a single GPU (Xeon Gold with RTX2080ti GPU) environment at present, we conducted a performance test using multiple GPUs (Xeon Gold with two RTX2080ti GPUs) on the same dataset, and we found that the execution time was significantly reduced to 39.2 min. This result showed that the multiple thread version of DARUMA is expected to enhance its usability, and it will be available in the near future.

## CONCLUSIONS

In this study, we developed DARUMA, a tool capable of delivering high-speed IDR predictions with accuracy comparable to that of state-of-the-art predictors. DARUMA is accessible *via* a web interface that accommodates both single protein queries and proteome-wide searches. Additionally, DARUMA is offered in several installation packages, enabling users to easily install the software in local environments. Users can benefit from high-speed predictions with DARUMA across a range of devices, from consumer-grade processors to sever-grade systems. DARUMA is accessible at https://antepontem.org/daruma/.

## ACKNOWLEDGEMENTS

The authors would like to thank Associate Professor Kenji Kawauchi for editing the English language of this manuscript.

### Funding

The authors received no funding for this work.

### Competing Interests

The authors declare that they have no competing interests.

### Author Contributions

- Itsuki Shimizu performed the experiments, analyzed the data, performed the computation work, prepared figures and/or tables, authored or reviewed drafts of the article, and approved the final draft.
- Takuya Ida performed the computation work, prepared figures and/or tables, and approved the final draft.
- Yuhei Ozawa performed the experiments, authored or reviewed drafts of the article, and approved the final draft.
- Satoshi Fukuchi conceived and designed the experiments, prepared figures and/or tables, authored or reviewed drafts of the article, and approved the final draft.
- Hiroto Anbo conceived and designed the experiments, performed the experiments, analyzed the data, prepared figures and/or tables, authored or reviewed drafts of the article, and approved the final draft.

## Data Availability

The data and code is available at GitHub and Zenodo:

- https://github.com/antepontem/DARUMA-Additional-Data.

- shimizuitsuki0316, & antepontem. (2025). antepontem/DARUMA-Additional-Data: DARUMA-Additional-Data (v1.1.0). Zenodo. https://doi.org/10.5281/zenodo.16564099.

The IDEAL dataset is available at: https://www.ideal-db.org/current.html.

The SL dataset is available at Bioinformatics Institute A*STAR Singapore: https://mendel.bii.a-star.edu.sg/SEQUENCES/disorder/.

The CAID1 dataset is available at CodeOcean: https://codeocean.com/capsule/2223745/tree/v1.

The CAID2 dataset and prediction results are available at: https://caid.idpcentral.org/challenge/results.

The RIDAO dataset is available at: https://ridao.app/publications.

The data and code are available in the Supplemental Files.

## Supplemental Information

Supplemental information for this article can be found online at http://dx.doi.org/10.7717/peerj-cs.3343#supplemental-information.

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
