# Peer review of "DARUMA: a gateway to fast and easy prediction of intrinsically disordered regions"

_PeerJ Computer Science, doi:10.7717/peerj-cs.3343_

## Round 0.1 · original submission · Major Revisions

· Academic Editor

Major Revisions

**Language Note:** When you prepare your next revision, please either (i) have a colleague who is proficient in English and familiar with the subject matter review your manuscript, or (ii) contact a professional editing service to review your manuscript. PeerJ can provide language editing services - you can contact us at [email protected] for pricing (be sure to provide your manuscript number and title). – PeerJ Staff

Reviewer 1 ·

Basic reporting

The manuscript is written using clear, unambiguous, professional, and technically correct English. The structure of the manuscript is in line with the acceptable format of a research article. Figures are relevant and of sufficient resolution.

The authors did not discuss that several convolutional neural network (CNN)-based predictors of protein intrinsic disorder had already been reported. In my view, at least some of those previously reported predictors should be compared with DARUMA.

A paper describing a high-efficiency web-based disorder predictor, Rapid Intrinsic Disorder Analysis Online (RIDAO), was published. In that paper, the efficiency and accessibility (i.e., ease of use) of RIDAO to five well-known and popular disorder predictors, namely: AUCpreD, IUPred3, metapredict V2, flDPnn, and SPOT-Disorder2, were analyzed. It was shown that RIDAO yields per-residue predictions at a rate two to six orders of magnitude greater than the other predictors. RIDAO operates in a single sequence and bunch modes. It has high efficiency and speed and can be used for the analysis of entire proteomes. In fact, it was shown that RIDAO can completely process the test set containing more than one million sequences from one hundred organisms comprising over 420 million residues in under an hour. Since the authors conducted similar analyses of their tool DARUMA, the paper talking about RIDAO should be cited and discussed.

Experimental design

Research question well defined, relevant & meaningful. It is stated how research fills an identified knowledge gap. Rigorous investigation performed to a high technical & ethical standard. Methods are described with sufficient detail & information to replicate.

However, the authors are encouraged to add analysis of some of the previously reported CNN-based disorder predictors and compare them with DARUMA.

The authors are reporting the efficiency of DARUMA in terms of the average execution time per protein across all proteins in the datasets. Since proteins have very different lengths, the better approach is to consider using the average execution time per residue.

Validity of the findings

All underlying data have been provided; they are robust, statistically sound, and controlled. Conclusions are stated well and supported by the results provided.

However, a comparison of DARUMA with already existing CNN-based disorder predictors is missing. The authors should also discuss RIDAO.

Cite this review as

Reviewer 2 ·

Basic reporting

-

Experimental design

-

Validity of the findings

-

Cite this review as

Reviewer 3 ·

Basic reporting

-

Experimental design

By introducing DARUMA, a lightweight 1D-CNN model based on physicochemical properties, the authors present a faster alternative to several top-performing disorder predictors, achieving comparable accuracy on the DisProt646 and DisProt-PDB datasets. If I understand the architecture correctly from lines 165–182 and Figure 1, the convolutional filter operates only along the feature dimension, while sequence correlations are captured through a sliding window of 17 residues applied to the output of the PU layer. Could the authors clarify how terminal residues are handled to produce an output of length L? Specifically, how is the sliding window centered, and is padding used?

Validity of the findings

I appreciate that the manuscript presents controlled accuracy metrics, with relevant data provided to support reproducibility. While the reported computational efficiency is a key strength of the work, the presentation of average runtimes across proteins of varying lengths may be somewhat misleading, as model performance may scale non-linearly with sequence length depending on architectural design. For example, Figure 3 could be reformatted to illustrate runtime scaling as a function of sequence length.
In lines 250-258, it's unclear to me if any parallelization is used in the DARUMA multi, or if the reported speed-up is solely due to avoiding model re-initialization. Thus, it's hard to interpret the multi runtime in context, since it is not clear whether equivalent settings were applied to the other methods if available.

Cite this review as

---

## Round 0.2 · Minor Revisions

· Academic Editor

Minor Revisions

Reviewer 3 ·

Basic reporting

The authors have provided a more comprehensive discussion of the literature.

Experimental design

The author has addressed my comments on model architecture.

Validity of the findings

The update of plots and analysis on residue normalized execution time is helpful and clarifies the scaling behavior. I have only a minor question: why reports per 1000 residues, and how are sequences shorter than 1000 residues handled?

The authors have also addressed my original comment by clarifying the source of the DARUMA(multi)'s speed-up gains. The comparison with RIDAO, showing that parallelization contributes to further improvements, is appealing and can potentially make a stronger case for the other datasets discussed in the paper.

Cite this review as

---

## Round 0.3 · accepted · Accept

· Academic Editor

Accept

Dear Author,

Your paper has been revised. It has been accepted for publication in PeerJ Computer Science. Thank you for your fine contribution.

Reviewer 1 ·

Basic reporting

The manuscript is written well.

Experimental design

Experimental design is appropriate.

Validity of the findings

The reported tool is useful and will be used by many researchers.

Additional comments

All my critiques were completely addressed, and the manuscript was revised accordingly. I do not have new or additional concerns.

Cite this review as

Reviewer 3 ·

Basic reporting

-

Experimental design

-

Validity of the findings

The authors have addressed my questions.

Cite this review as